# Interplay of Oxidative Stress and Necrosis-like Cell Death in Cardiac Ischemia/Reperfusion Injury: A Focus on Necroptosis

**DOI:** 10.3390/biomedicines10010127

**Published:** 2022-01-07

**Authors:** Adriana Adameova, Csaba Horvath, Safa Abdul-Ghani, Zoltan V. Varga, M. Saadeh Suleiman, Naranjan S. Dhalla

**Affiliations:** 1Department of Pharmacology and Toxicology, Faculty of Pharmacy, Comenius University in Bratislava, 83232 Bratislava, Slovakia; horvath125@uniba.sk; 2Centre of Experimental Medicine, Institute for Heart Research, Slovak Academy of Sciences, 81438 Bratislava, Slovakia; 3Department of Physiology, Faculty of Medicine, Al-Quds University, Abu Dis P.O. Box 89, Palestine; salghani@staff.alquds.edu; 4HCEMM-SU Cardiometabolic Immunology Research Group, Department of Pharmacology and Pharmacotherapy, Semmelweis University, 1089 Budapest, Hungary; varga.zoltan@med.semmelweis-univ.hu; 5Faculty of Health Sciences, Bristol Heart Institute, The Bristol Medical School, University of Bristol, Bristol BS8 1TH, UK; M.S.Suleiman@bristol.ac.uk; 6Institute of Cardiovascular Sciences, St. Boniface Hospital Albrechtsen Research Center, 351 Tache Avenue, Winnipeg, MB R2H 2A6, Canada; nsdhalla@sbrc.ca; 7Department of Physiology and Pathophysiology, Max Rady College of Medicine, Faculty of Health Sciences, University of Manitoba, Winnipeg, MB R3E 0J9, Canada

**Keywords:** necroptosis, apoptosis, oxidative stress, nitrosative stress, myocardial infarction, heart failure

## Abstract

Extensive research work has been carried out to define the exact significance and contribution of regulated necrosis-like cell death program, such as necroptosis to cardiac ischemic injury. This cell damaging process plays a critical role in the pathomechanisms of myocardial infarction (MI) and post-infarction heart failure (HF). Accordingly, it has been documented that the modulation of key molecules of the canonical signaling pathway of necroptosis, involving receptor-interacting protein kinases (RIP1 and RIP3) as well as mixed lineage kinase domain-like pseudokinase (MLKL), elicit cardioprotective effects. This is evidenced by the reduction of the MI-induced infarct size, alleviation of myocardial dysfunction, and adverse cardiac remodeling. In addition to this molecular signaling of necroptosis, the non-canonical pathway, involving Ca^2+^/calmodulin-dependent protein kinase II (CaMKII)-mediated regulation of mitochondrial permeability transition pore (mPTP) opening, and phosphoglycerate mutase 5 (PGAM5)–dynamin-related protein 1 (Drp-1)-induced mitochondrial fission, has recently been linked to ischemic heart injury. Since MI and HF are characterized by an imbalance between reactive oxygen species production and degradation as well as the occurrence of necroptosis in the heart, it is likely that oxidative stress (OS) may be involved in the mechanisms of this cell death program for inducing cardiac damage. In this review, therefore, several observations from different studies are presented to support this paradigm linking cardiac OS, the canonical and non-canonical pathways of necroptosis, and ischemia-induced injury. It is concluded that a multiple therapeutic approach targeting some specific changes in OS and necroptosis may be beneficial in improving the treatment of ischemic heart disease.

## 1. Introduction

The pathogenesis of ischemic heart disease, which accounts for about 20% of all deaths in the European Union [1], is associated with cardiac cell death. During the exposure to ischemia, cardiomyocytes begin to die, and severe and sustained occlusion of coronary artery promotes irreversible loss of cardiac cells, leading to the development of myocardial infarction (MI). The only way to rescue ischemic myocardium from infarction is reperfusion for a timely restoration of the blood supply. However, reperfusion beyond a certain time period has been shown to cause additional irreversible injury exacerbating the infarct process (lethal reperfusion injury) and thus significantly contributes to the myocardial infarct development [2,3,4]. It has been indicated that infarct size is a critical determinant of the outcome and is closely associated with all-cause mortality and hospitalization for post-ischemic heart failure (HF) within the following year [5]. Thus, death of cardiac cells occurring both immediately upon the deleterious ischemic impulse and in a later phase significantly determines the destiny of the heart, as well as the lifespan and the quality of life. Because cell death has also been documented in cardiomyopathies of various etiology, from diabetes to drug-induced pathologies [6,7,8,9,10], a similar pattern is likely to occur. Therefore, over the past few decades, enormous efforts have been made to understand cell death pathways with a special reference to programs that could be pharmacologically controlled. Indeed, apoptosis, which used to be considered as the only active, tightly regulated, genetically controlled, self-orchestrated cell death [11,12,13], has been targeted by various inhibitors of different caspases to limit infarct size [14,15,16]. However, due to improvements in laboratory techniques for accurate identification and measurement of apoptosis [17,18], the contribution of apoptosis to the extent of infarct size, as well as to the pathogenesis of ischemic heart disease and heart failure has been questioned. In fact, many studies dealing with acute or chronic ischemic heart damage have indicated only a minor role of apoptosis in such injury [10,19,20,21,22]. Likewise, the level of necrosis has been shown to be sevenfold 7-fold greater than apoptosis in patients with ischemic cardiomyopathy or idiopathic dilated cardiomyopathy [23]. In this regard, it should be noted that the majority of investigations in the field of myocardial apoptosis disregarded the occurrence of apoptosis in non-myocytes [24]. Indeed, it has been shown that apoptosis in non-myocytes is about 8- to 9-fold greater than in cardiac myocytes; macrophages are likely to represent the largest fraction of apoptotic non-myocytes [25]. In addition, various factors such as the occurrence of apoptosis during ischemia and/or reperfusion as well as consideration of these changes as an adaptation phenomenon have also been addressed [26,27,28].

From the aforementioned information, it is apparent that the role of apoptosis in the pathogenesis of MI and HF is still a subject of an intensive debate and controversy and that no conclusive statement can be made regarding this issue at this time. On the other hand, the importance of other regulated cell death modes resembling some features of passive necrosis in cardiac damage has been highlighted in the beginning of the new millennium. Accordingly, different terms such as pyroptosis, necroptosis, NETosis, and ferroptosis for defining cellular death were introduced in 2001, 2005, 2007, and 2012, respectively [29,30,31,32]. In the context of MI and post-ischemic HF, these cell death programs produce deleterious cellular responses; however, it has not been clarified to which extent these contribute to the development of infarct size and what is the exact proportion of their occurrence in any particular cardiac disease. It has been suggested that in some cases, the number of cells dying due to these individual necrosis-like deaths (besides others such as autophagy-dependent cell death and parthanatos) may outnumber those due to apoptosis [10,19,20,33,34]. These pathways can be stand-alone cell death modalities, but can also co-exist independently or sequentially, suggesting that these modes of cellular death overlap and utilize common signaling components. For instance, receptor-interacting protein kinase 3 (RIP3) has been identified to function as a convergence point of multiple signaling pathways, including necroptosis, inflammation, and oxidative stress (OS) [35,36,37]. Thus, necroptosis can be closely associated with pyroptosis and can underlie, or at least in part, contribute to the inflammatory response in post-ischemic HF [19,22]. Likewise, it has been documented that necroptosis can operate with autophagy, and both scenarios namely necroptosis activation as a suppressor of autophagic flux and autophagy suppression as a promotor of necroptosis have been reported in cardiomyocytes [20,38,39]. In addition, necroptosis has also been reported to accompany ferroptosis, a cell death mode defined as simultaneously occurring and mutually amplifying accumulation of redox-active iron/glutathione depletion, and lipid peroxidation [32,40,41]. Therefore, it is likely that the ischemia-mediated damage of the heart can be a result of cell loss due to several cell death programs and that targeting of multiple mechanisms can be of greater efficacy in improving the outcome.

OS, documented at various levels (as the higher levels of reactive oxygen/nitrogen species, changes in anti-/pro-oxidant enzymes and their respective genes, as well as altered activity of anti-/pro-oxidant enzymes), has been recognized as a significant player in the pathomechanisms of MI and HF [42,43,44,45]. Various deleterious molecular effects of OS have been reported to underlie alterations in cardiac performance and morphology; these include mitochondrial dysfunction [46], development of Ca^2+^-handling abnormalities and intracellular Ca^2+^ overload [47,48], activation of metallomatrix proteases and degradation of the extracellular matrix proteins [49,50], and loss of cardiomyocytes due to apoptosis [51,52]. In addition to apoptotic cell death, ferroptosis also occurs as result of OS. This concept is consistent with findings documenting this cell death mode in the ischemia/reperfusion (I/R)-induced heart damage [53,54,55]. Similarly, H_2_O_2_-induced injury, an in vitro model mimicking I/R, has been shown to cause gasdermin D (GSDMD)-mediated pyroptosis in adult mice cardiomyocytes and macrophages [33,56]. Other studies in the field of pyroptosis have also indicated a role of OS in the signaling of this programmed necrosis-like cell death. In this regard, NADPH oxidase enzyme isoform 4 (NOX4) can be a critical source of cardiac OS in hypercholesterolemia [57], I/R injury [58], as well as in post-ischemic HF [59], and may contribute to inflammasome activation that can in turn culminate to pyroptosis [60,61]. It is pointed out that the experimental evidence for OS in the signaling of necroptosis in the heart under conditions of I/R is rather limited, although there are some indications supporting this assumption. In this review, therefore, it is planned to discuss the current knowledge on OS in necroptotic cell death program generally and to focus on the potential underlying mechanisms of OS in the operation of necroptosis under ischemic (and marginally in non-ischemic) regions in the heart. In addition, an attempt will be made to analyze observations from various experimental cardiac studies dealing with this paradigm to test the hypothesis whether a modulation of OS could be a part of a cardioprotective, anti-necroptotic strategy, in a manner similar to approaches targeting the key necroptotic molecules such as RIP1, RIP3, and mixed lineage kinase domain-like pseudokinase (MLKL).

## 2. Mechanisms of Necroptosis Induction and Execution

### 2.1. Canonical Signaling Pathway of Necroptosis

The molecular mechanisms of necroptosis have been investigated previously and the proposed necroptotic pathways activated by ischemia in the heart have been reviewed elsewhere [62,63,64]. Briefly, necroptosis induction was found to be mediated by the stimulation of death receptors, such as TNFR1, FasR, TNF-related apoptosis-inducing ligand receptor (TRAIL-R), and T cell receptor [65]. The stimulation of such receptors appears to promote different executing processes of necroptosis involving the formation of necrosome or ripoptosome [66,67,68,69]. The TNFR1-mediated necroptotic pathway, which has been the most studied mode of necroptosis induction, involves necrosome assembly due to the autophosphorylation of RIP1 (pSer161) to further promote the phosphorylation of RIP3 (hRIP3 at Ser227, Thr231/Ser232 in mRIP3) [68,70]. As a result, RIP3 activates an enzymatically inactive hMLKL protein at Thr357 and Ser358 (at Ser358 in mMLKL) to form RIP3-MLKL hetero-oligomers, or MLKL homo-oligomers (tetramers and/or octamers) with resultant destruction of cellular integrity [71,72,73]. Several deleterious mechanisms inducing cell death due to the MLKL oligomers have been proposed. The cytoplasmic membrane rupture as a consequence of alterations in ion homeostasis leads to oncosis [74,75] and the attraction of proteases of the disintegrins and metalloproteinases (ADAMs) family inducing shedding of the diverse cell surface proteins, including receptors, adhesion molecules, growth factors, and cytokines [76]. In addition, translocation of MLKL within the plasma membrane has been shown to trigger activation of NACHT, LRR, and PYD domains, containing protein 3 (NLRP3) inflammasome complex resulting in the engagement of caspase-1 and subsequent interleukin-1β (IL-1β) cleavage before cell lysis [77]. With respect to the canonical signaling of necroptosis, it can also be mentioned that RIP1 seems to be dispensable for necroptosis induction [78], and that RIP3-RIP3 interaction, not RIP1-RIP3 heterodimeric amyloid fibril, is required for necroptosis. The RIP3-RIP3 dimerization is sufficient to induce this cell death by the recruitment of MLKL [78], arguing that RIP3 and MLKL are the core components of necroptotic signaling pathway. Thus, the analysis of these particular proteins (mainly their post-translational changes and RIP3-MLKL interaction as examined by immunofluorescence or by co-immunoprecipitation) along with other less specific analysis, for components such as lactate dehydrogenase (LDH) release, high mobility group box 1 (HMGB1) levels, annexin V/PI staining, indicating the cell membrane disruption, have been considered to provide proof for the occurrence of necroptotic cell death in the heart [17].

### 2.2. Non-Canonical Signaling Pathway of Necroptosis

In addition to the above-discussed RIP3-MLKL signaling pathway of necroptosis resulting in the plasma membrane rupture, other axes being associated with mitochondria have been indicated to promote necroptotic cell death in the heart [64]. One of the first identified proteins that has extended the canonical theory on the cell integrity disruption due to necroptosis was calcium/calmodulin-dependent protein kinase II (CaMKII). Overactivation of this multifunctional serine/threonine protein kinase by phosphorylation of threonine residue (Thr286/287) has been shown to contribute to the disturbances in ion homeostasis as well as inflammatory and apoptotic cell death signaling in myocardial I/R injury [79,80,81,82]. Experimental evidence from our laboratory has indicated for the first time that a pharmacologic inhibition of CaMKII is able to suppress the increased expression of RIP1 in acute myocardial I/R injury, thereby providing evidence for a possible link between these two kinases and the associated molecular events [82]. The identification of a role of CaMKII in necroptosis signaling has been suggested by showing that RIP3 may be its upstream activator. In fact, RIP3-mediated phosphorylation of CaMKII promoted mPTP opening with resultant mitochondria-mediated necroptosis execution in both I/R and in a model of doxorubicin-induced cardiac damage [35]. The recent findings that CaMKII may act as a downstream effector for MLKL [83] are fully in accordance with the proposed theory linking the canonical pathway of necroptosis with the non-canonical one and thus, may pave the way for the discovery of a novel CaMKII-associated necroptotic executive mechanism.

In the context of mitochondria-related signaling of necroptosis, another link involving phosphoglycerate mutase family member 5 (PGAM5), a mitochondrial outer-membrane serine/threonine-protein phosphatase, and dynamin-related protein 1 (Drp-1), a core component of mitochondrial division, has been delineated. In cardiac injury, PGAM5 expression has been shown to be elevated as it promotes dephosphorylation of Drp-1 on Ser637 which in turn leads in mitochondrial fission, elevation of reactive oxygen species (ROS), and finally cell death [84]. Consistent with a previous report on Drp-1 activation due to dephosphorylation on Ser637, [84], this signaling is activated upon RIP3 translocation to the mitochondria, and results in mitochondrial fragmentation and necroptosis activation [85]. Importantly, these molecular events occur independently of RIP1 and MLKL [85], thereby highlighting that RIP3 can be considered as a core mediator in necroptotic cell death. In agreement with this report, the inhibition of PGAM5 was reported to decrease the expression of necroptotic markers RIP1, RIP3, MLKL, and pSer616-Drp-1 in a model of in vivo myocardial I/R injury [86]. In addition to these findings arguing for pro-necroptotic RIP3–PGAM5–Drp-1 axis, several investigators have challenged its involvement in necroptosis execution, and proposed that both of these mitochondrial proteins are rather dispensable for necroptosis [87,88]. Likewise, Horvath et al. [37] have reported that RIP3 inhibition does not alter PGAM5–Drp-1 signaling in the hearts subjected to a 10-min reperfusion. Thus, this non-canonical necroptotic signaling seems to be more complex and requires further investigation.

A non-canonical signaling pathway involving c-Jun N-terminal kinase (JNK) and Bcl2 interacting protein 3 (BNIP3) has been proposed to execute necroptosis [89]. In H9c2 cells under hypoxic conditions, RIP3 deletion caused a downregulation of JNK and decreased BNIP3 translocation to the mitochondria, leading to sustained mitochondrial bioenergetics and reduced activity of LDH [17,89]. On the other hand, this signaling pathway was unlikely activated in an ex vivo model of acute myocardial I/R injury [37]. Although non-canonical signaling pathways of necroptosis have mainly been limited to mitochondria, there is also an indication for a role of the sarcoplasmic reticulum (SR) in this cell death. Indeed, Zhu et al. [36] have proposed that I/R injury-induced activation of RIP3 may elevate SR stress leading to Ca^2+^ overload and xanthin oxidase (XO)-dependent opening of mPTP. The canonical and non-canonical signaling pathways of necroptosis with respect to the subcellular organelle involved in such signaling axes are listed in Table 1.

## 3. General Role of OS in Necroptosis Signaling

Several studies have indicated a link between the changes in the main necroptotic proteins and OS; however, the underlying mechanism of how ROS overproduction is involved in this cell death program is unknown. It was shown that TNF-induced ROS production in NIH 3T3 cells is RIP3 dependent [90], and that ROS may act in a positive feedback loop to facilitate necrosome formation [91]. Besides RIP3, its upstream molecule RIP1 can also enhance ROS production. In fact, upon the treatment of L929 cells with TNF, mitochondrial ROS caused structural modification of RIP1, resulting in its autophosphorylation on Ser161 followed by RIP3 phosphorylation and formation of the necrosome [92]. Furthermore, it has been demonstrated that generation of ROS was suppressed in various RIP3- or MLKL-deficient cell lines [70,93,94], thereby providing robust evidence for the involvement of OS in the necroptotic signaling. Based on these findings it is likely that ROS act as a downstream effector of RIP1/3 and MLKL; this view is supported by the observations that gene silencing of NADPH oxidase 2 (NOX2), which is considered as one of the most important ROS supplies under pathological conditions [95], was protective against RIP3 overexpression-mediated necroptosis of cardiomyocytes as it improved the cell viability [35].

In view of a theory interlinking OS and necroptosis, it is noteworthy to mention the studies dealing with diabetes mellitus (DM), which is well known to be associated with an imbalance in redox system [96,97,98,99,100,101] as well as necroptotic injury as it is discussed below. In fact, it was proposed that necroptosis serves as a primary mechanism of islet cell death in the development of DM and during islet transplantation [102,103]. Moreover, the inhibition of necroptosis by necrostatin-1 (Nec-1) had similar protective effects on cognitive function and brain morphology as metformin in high fat diet (HFD)-induced prediabetes [104]. Similarly, a blockade of necroptosis prevented obesity-induced metabolic complications in various mice models of DM (HFD, ob/ob, db/db) [105]. On the other hand, it can be mentioned that high glucose-mediated cardiac injury seems to be mediated via the canonical necroptotic pathway since the levels of phosphorylated MLKL were unchanged in comparison to the normoglycemic group. Thus, non-canonical necroptosis signaling has been highlighted with a specific role of the activated CaMKII [106]. In addition to mediating insulin resistance due to some necroptosis signaling, OS also favors the occurrence of atherosclerotic damage. In fact, macrophages subjected to oxidized LDL were susceptible to die through RIP3-mediated necrosis [107] along with activation of pyroptotic pathways [108]. Further support comes from another study showing that atherogenic plaques have increased RIP3 and MLKL expression compared to healthy artery walls [109]. In summary, it is evident that OS, at least partially, plays a role in necroptosis, while its implementation in the pathomechanisms of this cell death program as well as the particular order of molecular events underlying the injury may be determined by several factors, such as the duration and type of specific pathological conditions as well as the particular cells being subjected to oxidative burst.

## 4. Possible Involvement of OS in Necroptotic Injury in the Heart

A search for the keywords ‘‘necroptosis” AND “oxidative stress” AND “heart’’ on MEDLINE (PubMed) at the time of writing this review retrieved 36 original articles. Studies proposing a direct/indirect link between OS, this programmed cell death mode, and cardiac dysfunction/cardiac cells damage were selected and are discussed below. In addition to these studies listed in Table 2 and to support the concept of a link between these events, we also present evidence on plausible antioxidant properties of some necroptosis inhibitors (Table 3). In this regard, it may be noted that several studies have reported the blockade of ROS produced by various stimuli, e.g., high glucose [106,110,111], H_2_O_2_ [112] and angiotensin II (Ang II) [113], which is accompanied by a decrease in necroptotic loss of cardiomyocytes. Other in vitro and in vivo analyses have also shown that pretreatment of H9c2 cells with N-acetylcysteine (NAC), an antioxidant drug, attenuated both doxorubicin- and high glucose-induced oxidative damage and at the same time inhibited necroptosis by downregulating the expression of its crucial modulator—RIP3 [110,114]. Furthermore, in a rat model of pressure overload-induced HF, NAC pretreatment alleviated left ventricle systolic dysfunction attenuated the increase in ROS, and mitigated necroptotic cell death activation as evidenced by the decreased levels of RIP1, RIP3, and MLKL [115]. Thus, these findings are fully consistent with our concept proposing a close relationship between redox state disturbances and necroptosis in the heart. Zhang et al. [39]. Proposed a direct link between I/R-mediated RIP3 activation and ROS production in isolated cardiomyocytes while the underlying mechanism was shown to include the activation of CaMKII with a subsequent increase in mPTP opening. In addition, these investigators indicated that RIP3 could also be connected to several other OS-modulators including NOX2, glycogen phosphorylase (PYGL), glutamate-ammonia ligase (GLUL), and glutamate dehydrogenase 1 (GLUD1), since knockout/silencing of the genes encoding these enzymes alleviated RIP3-mediated ROS increase [35]. Besides CaMKII, I/R-induced RIP3 activation also elevates SR stress, which in turn causes XO-mediated increase in ROS production, and finally mPTP opening and necroptotic death of cardiomyocytes [36]. In support, our recent experimental evidence in ex vivo model of myocardial I/R has also indicated XO-related redox state alteration by RIP3 [37]. In fact, RIP3 inhibition modulated the expression of XO and ameliorated I/R-induced calcium-sensitive mPTP opening [37]. Based on this observation, it is very likely that OS in the heart under various pathologic conditions is involved in the pathomechanisms of necroptosis and is linked to/regulated by RIP3, potentially via CaMKII- and XO-mediated increase in ROS production (Figure 1). However, the role of ROS or the co-play of the vicious ROS-induced ROS production in promoting necroptotic loss of cardiomyocytes cannot be ruled out.

## 5. Anti-Necroptotic Agents: Pharmacodynamic Features and Modulation of Oxidative Stress

Several pharmacological approaches to modulate the necroptotic signaling pathways have been recognized and different inhibitors of RIP1, RIP3, and MLKL have been developed. The particular mechanism of action of anti-necroptotic effects of these inhibitors is agent-specific and may include the inhibition of the kinase domain of the protein kinase (e.g., RIP1, RIP3 inhibitors) and/or the inhibition of the translocation of the necroptotic oligomers within the plasma membrane (e.g., necrosulfonamide—a MLKL inhibitor). In addition, some molecules elicit a dual mechanism (e.g., GSK’074—a RIP1/RIP3 dual inhibitor) while other agents block necroptosis by so far not precisely defined mechanisms. The anti-necroptotic agents RIP1 inhibitors, called necrostatins, are compounds of diverse chemical structure. Nec-1 was identified in 2005 as an allosteric inhibitor of RIP1 kinase activity [122] and has been widely used in preclinical cardiac studies on necroptosis. It was shown to reduce infarct size, ameliorate contractile dysfunction, and prevent adverse remodeling in models of acute MI and post-myocardial infarction HF [22,118,119,123,124]. However, because Nec-1 is also able to block indoleamine-2,3-dioxygenase (IDO) [125], an enzyme associated with inflammation, immunotolerance, and sensitizing the tumor to cell death [126,127], indicating its plausible ability to interfere with inflammation caused by ischemia, critical issues concerning its in vivo use in the field on necroptosis are emerging. In addition, Nec-1 has been shown to elicit pleiotropic action as evidenced by the modulation of protein machinery of excitation-contraction cycling and the transient increase in blood pressure in healthy rats without any necroptosis-promoting conditions [121,128]. In contrast, necrostatin-1s (Nec-1s), known as 7-Cl-O-Nec-1 analogue, does not target IDO but possesses the improved pharmacokinetic properties, such as superior potency in blocking RIP1 (IC_50_ = 210 nM vs. 494 nM for Nec-1) and higher metabolic stability [30,129,130]. In addition to Nec-1, Nec-1s has also been reported to reduce infarct size, ameliorate cardiac dysfunction as well as occurrence of arrhythmias and severity in a model of acute I/R injury [21]. Both Nec-1 and Nec-1s inhibit the death domain of RIP1 [30,122,131]. Another RIP1 inhibitor, GSK’772, which binds in an allosteric pocket of the RIP1 kinase domain, possesses an exquisite kinase specificity and is a potential candidate for treatment of inflammatory diseases, such as psoriasis, rheumatoid arthritis, and ulcerative colitis [132,133,134]. On the other hand, RIP3 kinase inhibitors, such as GSK’872, HS-1371, GSK’843, and GSK’074 [135,136], may be appropriate for studying necroptosis with a preference over RIP1 inhibitors because RIP3 functions as a downstream effector of RIP1 thereby targeting MLKL, the terminal executor of necroptosis, with a greater selectivity and potency. Recently, it has been documented that GSK’872 and HS-1371 significantly reduce LDH release due to acute myocardial I/R injury and prevent mPTP opening without affecting the canonical necroptotic pathway [37]. The anticancer drug B-Raf(V600E) inhibitor, dabrafenib is also considered to be a RIP3 inhibitor because it has been shown to competitively inhibit ATP binding with the RIP3 enzyme and decrease in RIP3-mediated phosphorylation of MLKL at Ser358 as well as disruption of the interaction between RIP3 and MLKL [137]. In ischemic brain injury, this agent reduced the infarct lesion size [138]. Although according to our best knowledge dabrafenib has not been used in any cardiac study so far, a similar pattern of findings can also be expected in a model of MI. Other anticancer drugs, such as ponatinib and pazopanib have also been shown to exert inhibitory action on RIP1 and RIP3. The former agent inhibits both RIP1 and RIP3, while the latter one preferentially targets RIP1 [139,140]. It is also pointed out that necrosulfonamide, an MLKL inhibitor, can be seen to modulate necroptosis as it blocks MLKL by targeting Cys86 located in α-helix 4. Other MLKL inhibitors, the xanthine class, also covalently bind to Cys86 within the N-terminal executioner domain, but the subsequent biochemical changes in MLKL are different. Necrosulfonamide locks the executioner and the pseudokinase domain in an inactive domain–domain orientation, while the xanthine class strengthens the inhibitory effect of α-helix 6 via interaction with Phe148 [141].

In addition to the main mechanism of action of anti-necroptotic agents discussed above, it is likely that these drugs modulate OS directly, or affect oxidative burst due to the retardation of necroptosis indirectly. In fact, Nec-1 has been reported to alter the gene expression of some OS markers as it increased NO synthase 2 (NOS2), cyclooxygenase-2 (COX-2), GRB2 associated binding protein 1 (GAB1), and glutathione peroxidase 1 (GPX1) as well as decreased cytochrome B-245 alpha chain (CYBA) and thioredoxin interacting protein (TXNIP) mRNA levels after left anterior descending artery (LAD) ligation in mice [118]. In another study employing a left circumflex coronary artery (LCx) ligation in pigs, having a higher translational potential for human studies, Koudstaal et al. [119] showed that Nec-1 is able to suppress nuclear ROS levels in the myocardium. Furthermore, Nec-1 was found to prevent the accumulation of ROS caused by paraquat intoxication [120]. Other RIP1 inhibitors of necrostatins class have also demonstrated the ability to modulate oxidative and nitrosative stress. In this regard, it can be mentioned that such pharmacological effects have been observed under non-cell death-related conditions and were accompanied by the alterations in the activation of some excitation-contraction cycling proteins [121]. These findings support a link between contractile function and redox state of the heart and compromised heart function under conditions of oxidative damage [142,143,144]. On the other hand, GSK’772, a novel RIP1 inhibitor, lacked such nitrosative stress-related effects, thereby indicating that such modulation is rather a drug class-specific feature [121]. RIP3 inhibition by GSK’872 has also been suggested to modulate OS as evidenced by a slight decrease in the expression of XO and significant decrease in MnSOD levels in the settings of acute myocardial I/R injury [37]. To the best of our knowledge, despite the depicted potential of RIP3 in altering the redox state of the cell, this is the only study indicating a plausible direct regulatory effect of a RIP3 inhibitor on OS. Similarly, we are not aware of any study with a MLKL inhibitor showing the ability for OS modulation.

## 6. Concluding Remarks

While the role of apoptosis in the pathogenesis of ischemia-induced cardiac damage has not been fully established it has become evident that non-apoptotic cell death modalities manifesting with necrotic morphology, such as necroptosis, ferroptosis, and pyroptosis, are more important than apoptosis itself. These modalities can operate standalone or in the cooperation with other cell death programs and thereby serve in the development of infarct size and worsening heart function as well as cardiac remodeling due to ischemia. These phenotypes of ischemic injury have been viewed to occur as a result of OS, and there is evidence indicating altered production/degradation of ROS during necroptosis. Accordingly, a hypothesis addressing a paradigm linking OS, necroptosis signaling, and cell injury has been formulated as a strategy to combat ischemic heart disease. Although particular cellular and molecular mechanisms have been discussed in detail, it cannot be concluded whether OS itself triggers necroptosis, or whether it is an individual consequence of ischemia-associated injury, thereby coexisting independently of necroptosis for producing final cardiac cell loss. Future extensive research work will be required for the solution to this puzzle and for clarification whether multiple approaches involving anti-necroptotic therapy and the modulation of OS can be a part of a cardioprotective strategy.

## Figures and Tables

**Figure 1 biomedicines-10-00127-f001:**
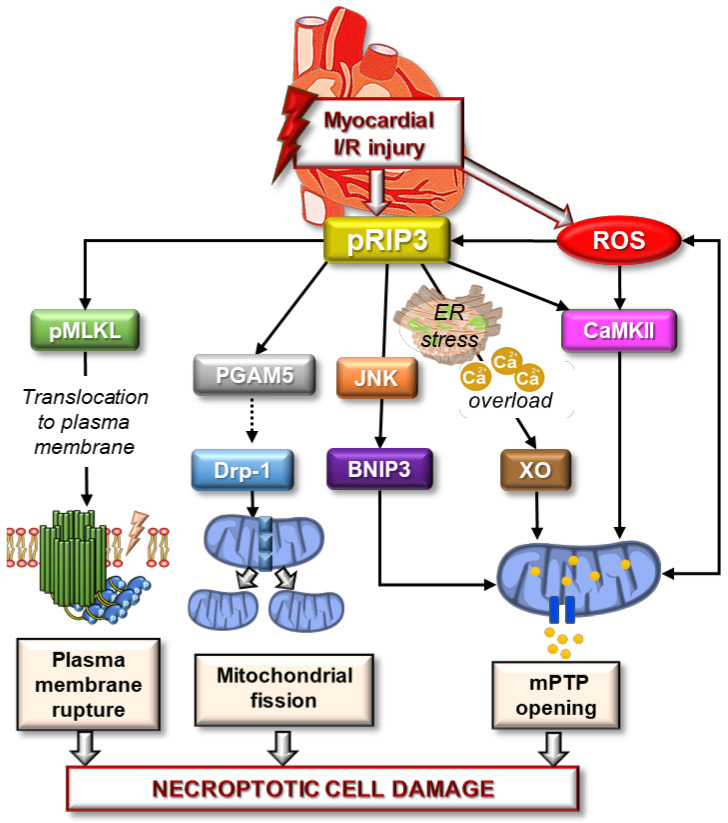
Schematic picture of molecular events indicating necroptotic cell damage via its canonical and noncanonical pathways under the conditions of myocardial ischemia/reperfusion injury. Myocardial ischemia/reperfusion injury leads to the activation of RIP3 promoting phosphorylation and subsequent translocation of MLKL to the plasma membrane causing its rupture. This canonical signaling pathway of necroptosis can also be activated by ROS which at the same time can promote the activation of CaMKII with resultant mPTP opening. In addition to the plasma membrane damage, RIP3 can advance necroptotic cell death under conditions of I/R via mitochondria-associated pathways. Firstly, RIP3 can activate PGAM5 dephosphorylating Drp-1 which causes mitochondrial fission. Secondly, it activates the protein kinases, such as CaMKII and JNK promoting mPTP opening. The induction of mPTP opening is also supported by RIP3-mediated Ca^2+^ overload and activation of XO. As a result of such mPTP opening, further ROS are produced thereby indicating a viscous cycle promoting other molecular pro-necroptotic RIP3-linked, both canonical and noncanonical, events. (I/R—ischemia/reperfusion, ROS—reactive oxygen species, RIP3—receptor-interacting protein kinase 3, mPTP—mitochondrial permeability transition pore, MLKL—mixed lineage kinase domain-like pseudokinase, PGAM5—phosphoglycerate mutase 3, Drp-1—dynamin-related protein 1, JNK—c-Jun N-terminal kinase, BNIP3—Bcl2 interacting protein 3, ER—endoplasmic reticulum, XO—xanthin oxidase, CaMKII—Ca/calmodulin-dependent protein kinase II).

**Table 1 biomedicines-10-00127-t001:** The proposed canonical and non-canonical signaling pathways of necroptosis.

Signaling Type		Proteins Involved	Reference
Canonical		RIP3–MLKL	[73,74,90,91]
Non-canonical	Associated with the mitochondria	RIP3–CaMKII	[35]
		RIP3–PGAM5–Drp-1	[85,86]
		RIP3–JNK–BNIP3	[89]
	Associated with the sarcoplasmic reticulum	RIP3–XO	[36]

**Table 2 biomedicines-10-00127-t002:** A list of studies showing a link between oxidative stress and necroptosis.

Model of OS-Induced Injury		Main Findings	Reference
Protocol	Cell/Animals		
30-min ischemia in vivo	Isolated adult mouse cardiomyocytes	ROS scavenger (Tiron) and knockdown of Nox2 mitigated RIP3-induced necroptosis	[35]
High glucose (25.5 mM)	Neonatal rat ventricular myocytes	I1PP1 overexpression decreased oxCaMKII and ROS levels and limited necroptosis	[106]
High glucose (35 mM)	H9c2	K_ATP_ channel opening was protective against high glucose-induced injury by inhibiting ROS-TLR4-necroptosis pathway	[111]
12-/24-/48-h hypoxia	H9c2	Pigment epithelium-derived factor ameliorated hypoxia-induced necroptosis and apoptosis activation via its antioxidant effect	[116]
H_2_O_2_ (500 μM)	H9c2	Dexmedetomidine prevents OS-induced necroptosis	[112]
Doxorubicin (1 μM for 24 h)	H9c2	NAC pre-treatment attenuated necroptosis by downregulating RIP3 and CaMKII expression	[114]
45-min LAD ligation	C57BL/6 mice	RIP3 mediates I/R injury via SR stress-Ca^2+^ overload-XO-ROS-mPTP pathway	[36]
Streptozocin-induced diabetes mellitus	C57BL/6 mice	Sirtuin 3 deficiency increased ROS production and promoted necroptosis	[117]
Abdominal aortic constriction	Rats	NAC (500 mg/kg) treatment prevented the increase in OS and necroptosis and improved LV systolic function	[115]

**Table 3 biomedicines-10-00127-t003:** List of studies illustrating an ability of necroptosis inhibitors to modulate oxidative stress.

Conditions	Experimental Protocol		Findings	Reference
**with ischemic insult**	30-min LAD ligation in vivo	Mice	Nec-1 (3.3 mg/kg) altered gene expression of NOS2, COX-2, GAB1 GPX1, CYBA and TXNIP	[118]
	30-min global ischemia ex vivo	Rat	GSK’872 (250 nM) decreased the expression of XO and MnSOD	[37]
	75-min LCx ligation in vivo	Pigs	Nec-1 (1.0 mg/kg; 3.3 mg/kg) decreased nuclear ROS levels	[119]
**without ischemic insult**	Angiotensin II (10 nM)	H9c2	Nec-1 (10 mM) reduced angiotensin II-induced ROS production	[113]
	High glucose (35 mM)	H9c2	Nec-1 (100 µM) reduced high glucose-induced ROS production	[110]
	Paraquat (45 mg/kg) administration in vivo	Mice	Nec-1 (3.5 mg/kg) reversed paraquat-induced ROS production in the heart	[120]
	Ex vivo perfusion	Rat	Nec-1 (1.2 µM), Nec-1i (1.2 µM) and Nec-1s (0.5 µM), but not GSK’772 increased protein tyrosine nitration	[121]

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
