# Peer review of "Interplay of Oxidative Stress and Necrosis-like Cell Death in Cardiac Ischemia/Reperfusion Injury: A Focus on Necroptosis"

_biomedicines, 2022, doi:10.3390/biomedicines10010127_

Round 1

Reviewer 1 Report

The review by Adameova et al., provides a concise summary of the existing literature linking oxidative stress and necroptosis in the context of cardiac ischemic injury. In general, the authors have done a good job. The primary literature is well cited, and the manuscript is well organized overall. However, moderate English language editing is required to make the review more readable. Nonetheless, I am confidant that with additional English editing, this manuscript will be acceptable for publication.

Author Response

Dear Editors-in-Chief,

Thank you for your email dated December 29, 2021. We would also like to thank the reviewers for their valuable comments and for their interest in our work. As you can see below, all the comments have been addressed and the particular parts of the main text have been revised (by using Track Changes function).

We believe that the revision has improved the quality of our article and hope it will meet with your approval.

Kind regards,

prof. PharmDr. Adriana Duris Adameova, PhD.

Reviewer #1

The review by Adameova et al., provides a concise summary of the existing literature linking oxidative stress and necroptosis in the context of cardiac ischemic injury. In general, the authors have done a good job. The primary literature is well cited, and the manuscript is well organized overall. However, moderate English language editing is required to make the review more readable. Nonetheless, I am confidant that with additional English editing, this manuscript will be acceptable for publication.

The manuscript has underwent a careful language editing provided by native English speaking co-authors.

Reviewer 2 Report

In this manuscript, the authors reviewed the available studies that support the possible linkages between cardiac oxidative stress and necroptosis in I/R injury and heart failure. This review is well-written and will be very helpful to the field.

Here are some points the authors may want to address:

  1. At the whole review level. The authors spent too many efforts on introduction parts: parts 1, 2 and 3. It’s until parts 4, the authors started to discuss the possible relations among OS, necroptosis and cardiac I/R injury. The authors may want to shorten/merge parts 1, 2 and 3, and further expand parts 4 and 5.
  2. Some of these references are either outdated or mismatched, the authors may want to further double check if all these references are cited properly. Here are some examples:
    1. Reference 1, the census was published in 2014. There are new stats published in recent years. Here is an example: European Society of Cardiology: Cardiovascular Disease Statistics 2021
    2. References 2-5 could be condensed further.
    3. Line 58, references 11 does not match the statement here.
    4. Line 67-68, reference 20. The original paper found that “necrosis is 7-fold greater than apoptosis”, which is different from the authors’ statement here “apoptosis is 7-fold lower than necrosis”. The authors may want to rewrite this sentence like “apoptosis is only XX% of necrosis” to make it straight forward.
    5. Line 69, reference [11, 21-24], these references do not support the statement here.
    6. Line 83-84, references [30-33] were published in 2005, 2012, 2005 and 2007, which do not match the time the authors stated here as 2001, 2005, 2007 and 2012.

Author Response

Dear Editors-in-Chief,

Thank you for your email dated December 29, 2021. We would also like to thank the reviewers for their valuable comments and for their interest in our work. As you can see below, all the comments have been addressed and the particular parts of the main text have been revised (by using Track Changes function).

We believe that the revision has improved the quality of our article and hope it will meet with your approval.

Kind regards,

prof. PharmDr. Adriana Duris Adameova, PhD.

Reviewer #2

  1. At the whole review level. The authors spent too many efforts on introduction parts: parts 1, 2 and 3. It’s until parts 4, the authors started to discuss the possible relations among OS, necroptosis and cardiac I/R injury. The authors may want to shorten/merge parts 1, 2 and 3, and further expand parts 4 and 5.

As it is indicated in the very first sections of Introduction, the current knowledge on oxidative stress in necroptotic cell death program has been discussed generally and a particular focus on the potential underlying mechanisms of oxidative stress in the operation of necroptosis under ischemic (and marginally in non-ischemic) regions in the heart has been addressed. In addition, an attempt was made to analyze observations from various experimental cardiac studies dealing with this paradigm. Because OS is associated with diabetes and atherosclerosis, in which necroptosis has been identified as an important pathomechanism of these diseases, we prefer not to shorten these sections (Introduction part 3) to provide the indirect indication of a link between oxidative stress and necroptosis. Furthermore, because there is a limited number of studies with a proven link between oxidative stress, necroptosis and cardiac damage under conditions of ischemia, the section 4 can be seemingly shorter than it would be expected. On the other hand, in our opinion, the section 5 has adequately provided a concise summary of agents being developed to target the main molecules of necroptotic signaling and in line with the hypothesis has also indicated their capability of modulating oxidative stress.  

All these sections have been carefully read and some revision has been done as advised.

Some of these references are either outdated or mismatched, the authors may want to further double check if all these references are cited properly. Here are some examples:

Reference 1, the census was published in 2014. There are new stats published in recent years. Here is an example: European Society of Cardiology: Cardiovascular Disease Statistics 2021

In order to update the disease statistics, the reference published in 2014 has been replaced by a paper of the ESC Atlas of Cardiology from 2021. In this manner we would like to indicate that the referred paper on the updated disease statistics has not been available at the time of submitting our manuscript. Indeed, „European Society of Cardiology: Cardiovascular Disease Statistics 2021“ has been published on 23 December 2021 while our manuscript was submitted on 1 December 2021 [1].

References 2-5 could be condensed further.

In that particular part, one reference has been deleted and careful check of other references has been done throughout the entire manuscript and their revision has been done when needed. For instance, the duplicated reference on Guidelines for evaluating myocardial cell death, and the ref. on Li et al. 2017 has been removed, etc.

Line 58, references 11 does not match the statement here.

We do not agree that Ref. No 11, in this version it is Ref. No 10 (Szobi et al. 2017) does not match the statement. The paper deals with necroptosis in human heart failure of various etiology so it perfectly refers to the first part of the sentence „Because cell death has also been documented in cardiomyopathies of various etiology,...“. However, to be more clear we have modified this sentence. Please see p. 58 and 59.

Line 67-68, reference 20. The original paper found that “necrosis is 7-fold greater than apoptosis”, which is different from the authors’ statement here “apoptosis is 7-fold lower than necrosis”. The authors may want to rewrite this sentence like “apoptosis is only XX% of necrosis” to make it straight forward.

We are of opinion that the statement „necrosis is 7-fold greater than apoptosis“ is equivalent to  „apoptosis is 7-fold lower than necrosis“. However, to avoid any possible misinterpretation, we decided to rephrase the particular sentences accordingly. Please see p. 63-70.

Line 69, reference [11, 21-24], these references do not support the statement here.

These references refer to a minor role of apoptosis in heart damage due to ischemia. For instance the first referred paper, an original paper from our laboratory, has clearly shown that necrosis-like cell death exceeds apoptosis in all investigated forms of heart failure, thereby arguing for a lower level of apoptosis in such conditions. Similarly, our other original papers (ref. No 21-23) and a paper of others (ref. No 24) have indicated a predominant role of necroptosis rather than apoptosis in hearts subjected to various ischemic attacks. Therefore, we do believe that these references support the statement on a minor role of apoptosis in heart damage and thus they have been kept. However, because of revision of the text along with the changes in some references, these references are numbered as Ref. No. 10, 19-22.

Line 83-84, references [30-33] were published in 2005, 2012, 2005 and 2007, which do not match the time the authors stated here as 2001, 2005, 2007 and 2012.

The only reference which could be considered inappropriate is the reference of Fink and Cookson, 2005. In this paper published in 2005 [2], the main author refers to his another paper published in 2001 [3] when he proposed the term pyroptosis to describe proinflammatory programmed cell death. This may be the reason of the interchangeable references. In order to provide the proper reference on the paper in which the term of pyroptosis was coined by Cookson and his coworkers, we have replaced the reference Fink and Cookson, 2005 by Cookson and Brennan, 2001. In addition, the order of the references has been modified accordingly.

  1. Group, A.W., et al., European Society of Cardiology: Cardiovascular Disease Statistics 2021. European Heart Journal, 2021.
  2. Fink, S.L. and B.T. Cookson, Apoptosis, pyroptosis, and necrosis: mechanistic description of dead and dying eukaryotic cells. Infect Immun, 2005. 73(4): p. 1907-16.
  3. Cookson, B.T. and M.A. Brennan, Pro-inflammatory programmed cell death. Trends Microbiol, 2001. 9(3): p. 113-4.